# Effects of Autologous Conditioned Serum on Non-Union After Open Reduction Internal Fixation Failure: A Case Series and Literature Review

**DOI:** 10.3390/medicina60111832

**Published:** 2024-11-07

**Authors:** Pen-Gang Cheng, Man-Kuan Au, Chian-Her Lee, Meng-Jen Huang, Kuender D. Yang, Chun-Sheng Hsu, Chi-Hui Wang

**Affiliations:** 1Department of Orthopedics, Fu-Ya Medical Clinic, Taichung 40764, Taiwan; 8104@ktgh.com.tw; 2Department of Orthopedics, Cheng-Hsin General Hospital, Taipei 11220, Taiwan; ch6200@chgh.org.tw; 3Department of Orthopedics, School of Medicine, College of Medicine, Taipei Medical University, Taipei 11031, Taiwan; lee060008@tmu.edu.tw; 4Department of Orthopedics, Taipei Medical University Hospital, Taipei 11031, Taiwan; 5Department of Orthopedics, Taipei Tzu-Chi Hospital, Taipei 23142, Taiwan; ortho_huang@tzuchi.com.tw; 6Mackay Children’s Hospital, Taipei 10449, Taiwan; yangkd.4499@mmh.org.tw; 7Department of Medical Research, Mackay Memorial Hospital, Taipei 10449, Taiwan; 8Department of Physical Medicine and Rehabilitation, Taichung Veterans General Hospital, Taichung 40705, Taiwan; 9Department of Post-Baccalaureate Medicine, College of Medicine, National Chung Hsing University, Taichung 40220, Taiwan; 10Department of Orthopedics, Cheng-Ching General Hospital, Taichung 40764, Taiwan; 15674@ccgh.com.tw

**Keywords:** autologous conditioned serum, fracture, non-union

## Abstract

*Background and Objectives*: Non-union is a severe complication of traumatic fracture that often leads to disability and decreased quality of life, with treatment remaining complex due to a lack of standardized protocols. This study examines the effectiveness of autologous conditioned serum (ACS) for non-union in patients who have a failed open reduction internal fixation (ORIF). *Materials and Methods*: Eleven patients with confirmed non-union at least 9 months post-ORIF or total hip replacement were enrolled. These patients received ACS treatment on the lesion sites once to three times monthly and were followed up. Efficacy was monitored through monthly X-rays to assess callus formation and bone union. *Results*: Seven patients received ACS three times, three patients received it twice, and the one who underwent total hip replacement received it once. Ten patients achieved union at the last follow-up visit, indicating the effectiveness of ACS in non-union cases unresponsive to ORIF. ACS demonstrated promising results in facilitating bone union in these challenging cases. *Conclusions*: ACS has the potential as an alternative or adjective treatment for non-union and is worthy of being investigated further for the benefits of patients.

## 1. Introduction

Non-union is one of the severe complications of traumatic fracture, which leads to functional impairment, morbidity, and loss of quality of life [1]. The incidence of non-union among studies varies, and a range from 2 to 15% has been reported, depends on the types of fracture and injured bone [2,3]. There are still no standard therapeutic recommendations for non-union. Local infection control, debridement, deformity correction, fixation, bone graft, and biological agents are all considered and utilized based on clinical experience and positive results gained [4]. Among these strategies, transplanting autologous cancellous graft on the non-union site might have the highest level of consensus as a more effective treatment. Autologous cancellous graft acts as a scaffold and source of bone cells for new bone growth. However, the limited source indicates the need for developing a new strategy. Platelet-rich plasma (PRP) is another biological agent that gains elevated attention for its potential role in orthopedic management, including non-union. Results of numerous basic, preclinical, and clinical studies have been published, and the efficacy of PRP alone or combination with other treatments on non-union has been proven [5,6,7]. Despite the majority of the literature supporting the positive effectiveness of PRP on non-union, conflicting results exist [8,9,10]. In addition, studies have shown that other biological agents may be more effective than PRP. Within the family of bone morphogenetic proteins (BMPs) that are released from mesenchymal stem cells and trigger chondroblastic and osteoblastic differentiation, BMP-7 shows superior efficacy during revision surgery compared to PRP [11,12]. BMP-2 is the first and only bioagent approved by the U.S. Food and Drug Administration for non-union treatment [13,14]. Furthermore, the lack of standardized preparation and administration protocols also hinders the establishment of consensus to include PRP in the recommendation for non-union treatment. Thus, alternative and adjective treatments for non-union are still being evaluated in the orthopedic field.

Autologous conditioned serum (ACS) is another autologous blood-derived product, which is composed of enriched cytokines and growth factors that are secreted by blood cells including platelets after being stimulated by glass beads. The first device developed for processing ACS was originally branded as “Orthokine” in the late 1990s [15,16,17]. Previous studies have shown the efficacy of ACS on osteoarthritis [15,16,18,19,20]. We have found that compared to PRP, ACS had a 4.7 times higher level of PDGF-BB (9512.7 vs. 2035.2 pg/mL) [21]. PDGF is a well-known important component during bone formation and regeneration by involving the vasculature–pericyte–mesenchymal stem cell–osteoblast dynamics. At the injury site, PDGF-BB promotes the mobilization of the pericytes from the abluminal location and stimulates mitotic expansion to achieve both new vasculature formation and osteoblast differentiation and osteogenesis for new bone formation in the fracture repair process [22]. To the best of our knowledge, there is, currently, no study about using ACS injection for patients with non-union fracture after open reduction internal fixation (ORIF). In this study, we attempted to evaluate the clinical efficacy of ACS injection to explore the possible clinical utility of ACS. The efficacy of ACS on non-union was observed by X-ray evaluation on follow-up visits.

## 2. Materials and Methods

### 2.1. Patients

Patients aged ≥ 18 years with fracture were eligible to receive the ACS treatment if it was radiologically confirmed by certified physicians that non-union was still present at least 9 months after ORIF [23]. They were asked to provide consent to participate in this study. In addition, one patient with a non-union anterosuperior bone defect at 25 months after total hip replacement was also included. Patients with hypotrophic non-union, fibrosis non-union, unstable fixation, open wounds, major neurological diseases such as dementia, active thrombovascular diseases, or infections, and those who were uncooperative were excluded. This study was approved by the Institutional Review Board of Kuang-Tien General Hospital (KTGH11105). All patients provided signed informed consent. From 2022 to 2024, a total of eleven patients were enrolled in this study.

### 2.2. ACS Preparation

ACS was prepared from 10 mL blood of each patient. Non-fasting blood samples were collected and transferred into an ACS sterile glass bead-containing tube (Pen-Ling Biotechnology Co., Ltd., Taichung, Taiwan), whose diameter was 16.2 mm and length was 129.2 mm, with one air pore on the top and medical-grade glass beads inside, and incubated at 37 °C for 3 h as described previously [21], followed by centrifugation at 4000 rpm for 5 min. ACS was harvested using a spinal needle through the air pore and transferred into a syringe for injection.

### 2.3. ACS Injection and Follow-Up

Patients were injected with 3.5 mL of ACS on the lesion under the guidance of ultrasound or C-arm fluoroscopy (Figure 1), and ACS was freshly prepared for each treatment appointment every month. Patients were followed up monthly. The clinical efficacy of the ACS injection on non-union was evaluated by X-ray at the next monthly visit. The number of ACS treatments, ranging between one and three, depended on the bone union status at the follow-up evaluated by the physicians. For example, if the X-ray showed the callus one month after the 2nd ACS injection, the scheduled 3rd injection was waived.

## 3. Results

### Patients and Outcomes of ACS Injection

From 2022 to 2024, a total of eleven patients were treated by ACS injection(s) on the lesions of non-union one to three times monthly and followed up (Table 1). Eight were males and three were females, aged between 21 and 66 years, and the median and mean were 45 and 43 years, respectively. Nine patients underwent surgical intervention due to long bone fractures, and the others underwent surgery because of hip osteoarthritis and clavicle fracture. Ten patients were treated by ORIF and one (Case 7) underwent a total hip replacement, but none could achieve bone union at least 9 months after initial surgical intervention. In one patient (Case 1), revisional intramedullary nail fixation was performed after plate implant failure; however, there was still no improvement. Four patients (Case 1, 2, 3, and 9) showed hypertrophic non-union and the others had bone defects. In terms of ACS treatment, seven patients received it three times, three patients received it twice, and the only patient who had undergone total hip replacement received one ACS injection. The time to first callus shown on X-ray images after the first ACS injection was from 1.5 to 5 months. Ten patients achieved union at the last follow-up (Cases 10 and 11 were followed up for 6 months, and the others were followed up for 12 months). As for the only patient (Case 5) who could not achieve bone union after ACS treatment, this was presumably due to implant failure caused by unstable fixation.

The radiographic results of representative cases, namely, Cases 1, 4, 6, and 11, are shown in Figure 2, Figure 3, Figure 4 and Figure 5, respectively. Case 1 was treated by plate implant (Figure 2a) and revisional intramedullary nail fixation (Figure 2b), but the fractural space still existed after 9 months (Figure 2c). Union was achieved at the last evaluation, which was three months after the third injection (Figure 2d).

Case 4 only received two ACS injections for non-union clavicle fracture, because the progress of bone healing was excellent and the physician decided that no more treatment was required (Figure 3). In the present study, up to three ACS injections were designed as the standard therapeutic plan based on clinical experience; however, the physicians could alter the plan according to clinical response. Thus, regular monitoring is necessary to make decisions. Radiography is an effective method with good availability for evaluating bone healing. Thus, routine radiographic evaluation was performed before and during ACS treatment in this study for the evaluation of the effectiveness of ACS.

Case 6 suffered from non-union femur fracture after ORIF for nine months, and then was treated by ACS injections three times, and complete bone healing was achieved (Figure 4).

Case 11 had both radial and ulnar fractures on the left arm, and received three ACS injections for non-union nine months after ORIF. The healing efficacy of ACS was observed from the results of the X-ray evaluation (Figure 5).

## 4. Discussion

Data regarding using ACS for fractures in addition to non-union treatments is limited; to date, the present study might be the first one that aims to evaluate the therapeutic efficacy of ACS on non-union. Before accumulating sufficient evidence of ACS by in vitro, in vivo, and clinical studies in the future, most of the perspectives of ACS in the study are derived from previous experiences and studies of PRP on bone healing.

The mechanism of PRP for treating non-union consists in providing concentrated platelet and specific growth factors, including platelet-derived growth factor (PDGF), vascular endothelial growth factor (VEGF), transforming growth factor (TGF), and insulin-like growth factor-1 (IGF-1) to the region of bone defect to establish the required microenvironment for bone regeneration. These effectors induce the process of bone healing, including cellular proliferation, matrix formation, osteoid production, and collagen synthesis that are achieved through activating and regulating osteoblasts, osteoclasts, and mesenchymal stem cells (MSCs), and are represented by soft callus formation, hard callus formation, and bone remodeling [7,24,25]. That means that both new skeletal tissue and surrounding soft tissue are reconstructed under the effects provided by PRP [25,26]. The plurality of involved growth factors, cells, and physical functions reflect the extraordinarily high complexity of the bone healing process, while autologous blood-derived products such as PRP or ACS, which are composed by many cytokines and growth factors, may have better and more comprehensive effects than one specific component since multiple pathways should be activated for initiating bone healing, and interaction between components may have synergy.

The efficacy of PRP on non-union has been proven by previous studies. A randomized, double-blinded, placebo-controlled study evaluated the efficacy of PRP in patients with non-union fracture together with autologous bone graft and internal fixation, with saline serving as the placebo. PRP showed a therapeutic effect on non-union, indicated by the significantly higher bone healing rate compared to that in the placebo group, 81.1% vs. 55.3% (*p* = 0.025) [27]. A meta-analysis including the above-mentioned clinical trial and another two randomized controlled studies, one prospective study, and nine retrospective studies showed similar results; compared to the control group, patients suffering from long-bone delayed union and non-union treated by PRP had a higher bone healing rate (85.8% vs. 60.27%; OR = 3.07; 95% CI, 1.37–6.87; *p* = 0.006) [28]. Using PRP alone or combined with other conventional treatments, i.e., autografts transplant, compression plates; and/or fixation devices, has been reported and compared by many investigators with various study designs [5]. Clinical trials were conducted for the comparison of PRP alone and combined with internal fixation. The combination of PRP and surgery provided a higher bone healing rate (94% vs. 78%; *p* < 0.05) and shorter bone healing time (91.6 ± 6.9 vs. 115.2 ± 8.4 days; *p* < 0.05) compared to those of PRP alone, indicating the benefit of incorporating autologous biologic agents into established clinical practice [5]. In addition, surgery combined with other biological agents such as MSCs for non-union was also evaluated [29,30], indicating that the unmet need is critical and more studies about newly developed biological agents should be performed to evaluate the effect(s) on new bone regeneration.

Even though ORIF is the standard treatment for fractures of numerous sites, internal fixation is sometimes challenging due to the complex anatomy, leading to the relatively high non-union rate despite the continuing advances in surgical technique [31]. The treatment of non-union after successful ORIF is also a challenge. For distal humerus fractures, up to 25% of non-union occurred after ORIF [32], and additional surgery including revision ORIF is the current recommendation [31,33]. However, the complication rate is very high with unsatisfactory outcomes, and bone healing may be not achieved after this secondary operation [31]. Thus, alternative treatment, such as applying biological agents to promote bone healing, still needs to be investigated for treating non-union after ORIF.

Many risk factors associated with non-union are known, including the location and type of bones. For example, as the weight-bearing bones, the fractures of the femur and the tibia usually need more time to reach union [34]. Increasing age, alcohol consumption, smoking, and nutritional deficiencies such as Vitamin D, calcium, or protein that are essential for bone healing are associated with delayed union or non-union [35]. In particular, the success rate of ORIF can be significantly impacted by the age of the patients. In elderly patients, because of the relatively high prevalence of osteoporosis and the poor condition of the surrounding soft tissue, the non-union rate may increase [36]. Three injections of ASC still could not achieve union in Case 5 and the possible reason was unstable fixation, which may be associated with the patient’s advanced age. Furthermore, from the experiences of PRP, the type of bone may also affect the healing outcomes and lead to controversial results in the literature; positive effects were often found in well-vascularized cancellous bone where cells for new bone regeneration were abundant, and cases of no effects usually had defects on the maxillary bone or the mandibular bone [37,38]. One patient (Case 7) who had undergone total hip replacement was also included in this study. Only one injection was needed to achieve union, and this may also indicate the impact of bone type on the efficacy of biological agents such as ACS.

Since the healing process and impact factor are both complicated, the optimalization of treatment plans that combine surgical intervention and autologous biological agents for individual needs should be considered based on the evidence. For example, the optimal timing of treating non-union fractures using autologous biological agents is still being evaluated. The experience from PRP shows that varied intervals after injury among studies, from six to twelve months, have been reported [39,40]. The present study defined non-union as at least nine months after the initial intervention for the treatment of injury, and the results indicated that ACS injection has a therapeutic effect on non-union fractures at least nine months after ORIF. Considering that the non-union rate in patients who have undergone the standard ORIF treatment is still high, ACS might be used earlier in the early phase of the fracture. Furthermore, the conventional treatments for fractures, including surgery, immobilization, and rehabilitation, are all time-consuming and significantly impact patients’ quality of life. If ACS is used as an alternative or adjective treatment, not only will it improve the bone healing rate and enhance the healed bone’s structural integrity but it will also shorten the time for bone healing, possibly greatly enhance the outcomes, and provide benefit on quality of life for patients suffering from fractures. The optimal number of injections is also unclear. In the case of using PRP for osteoarthritis, Subramanyam et al. [41] reported that three injections were suggested since superior outcomes to single and double injections were observed, and Andersen et al. [5] indicated that for non-union, usually a single dose was given, but in some studies, multiple injections were applied depending on the status of the fractural site [42,43]. In our previous study, PRP or ACS was given five times every 2 weeks for osteoarthritis [21]. In the present study, up to three injections were applied because postponing additional surgery until 12 months after injury may obtain unsatisfactory patient outcomes. The injection interval can be shortened in patients with a slow healing rate to increase the total dose of ACS, but the treatment protocol needs more studies to be established.

The effectiveness of PRP and ACS was not compared directly in the present study; however, from the study we conducted for osteoarthritis, ACS may be superior to PRP because of the higher levels of PDGF-BB and IL-1Ra in ACS. PDGF-BB and IL-1Ra levels were around 4.7 times (9512.7 vs. 2035.2 pg/mL) and 5.7 times (902.9 vs. 157.9 pg/mL) higher in ACS than in PRP [21]. PDGF-BB is involved in cell proliferation, migration, and angiogenesis. In the process of bone healing, PDGF-BB is responsible for inducing the chemotaxis of osteoblasts which are essential cells to initiate the synthesis of new bone tissue [44]. IL-1Ra is the antagonist of IL-1β, and serves as an immunomodulator during bone healing because of its anti-inflammatory properties. In the early phase of bone injury, proinflammatory cytokine IL-1β recruits inflammatory cells to debride the lesion and stimulate the transient matrix hematoma for the recruitment of mesenchymal stromal cells to rebuild new tissue [45]. However, prolonged inflammation hinders the bone healing process because of the inhibition of MSC differentiation and the induction of apoptosis [46]. The addition of IL-1Ra has been shown to enhance the healing of femoral defects in animal experiments [47]. Other components within these autologous biological agents probably also play an important role in the bone healing process, such as exosomes, which are considered capable of promoting tissue regeneration [47,48], and whose levels in ACS were 1.5 times higher than those in PRP (1.5 × 10^13^ ± 3.0 × 10^12^ vs. 6.0 × 10^12^ ± 4.9 × 10^11^ vesicles/mL). A detailed analysis of their contents and functions on bone healing should be conducted to explore the mechanism underlying their therapeutic effect.

ACS injection has another advantage that can be emphasized. Since ACS injection is a relatively simple procedure in clinical practice, robotic surgery or 3D printing might be not necessary. Thus, the utility range is speculated to be wider because of its simplicity, and ACS injection can be beneficial for more patients who suffer from non-union. The potential complications with ACS injection, however, should also be mentioned. Infection, bleeding, injection pain, nerve injury, or vessel injury may occur after ACS injection. Proper monitoring should be conducted in the clinical setting for these complications. The limitations of this study should be considered. The major limitation is the small sample size, so further studies with larger populations should be conducted to accumulate clinical experience. In addition, since the blood supply might be limited in fibrosis tissue, the efficacy in patients with fibrosis non-union is speculated to be lower, and such patients may be not suitable for ACS treatment before the fibrosis is removed. For the same consideration, the intramedullary cavity is a better site for ACS injection compared to the periosteum, and if possible, intramedullary injection should be chosen. From the results, the only patient who still had failed union had unstable fixation, and probably ACS treatment is unsuitable for this kind of patient. Given the fact that PRP in a gel formulation provided good benefits for patients with delayed union or non-union [39], further studies to make a gel formulation of ACS for patients with delayed union or non-union are needed.

## 5. Conclusions

In this retrospective study, the potential assistive role of ACS in bone healing in patients suffering from non-union after standard ORIF treatment was demonstrated. However, due to the small sample size and retrospective nature, prospective studies with larger study populations should be performed to provide more evidence for using ACS injection in clinical practice. In addition, the optimal timing, injection numbers, and total dose should be optimized after collecting more data and clinical experience from more patients with various conditions. Furthermore, the molecular mechanism and comparison with other autologous biological agents should be clarified in the future, in order to expand understanding and knowledge.

## Figures and Tables

**Figure 1 medicina-60-01832-f001:**
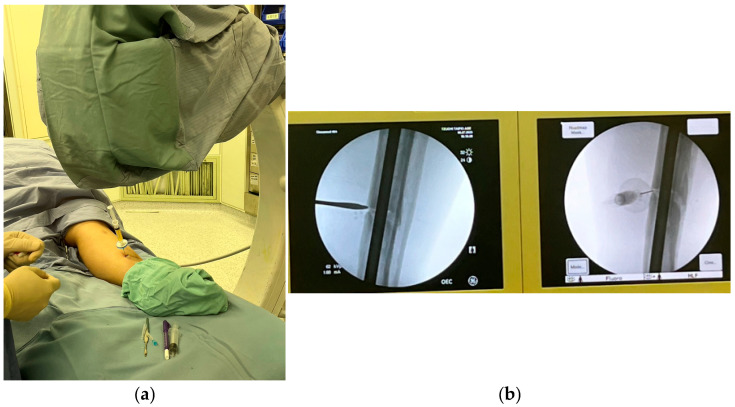
Case 1 was injected under the guidance of C-arm fluoroscopy. (**a**) The positioning of the patient; (**b**) the guidance to inject into the periosteum.

**Figure 2 medicina-60-01832-f002:**
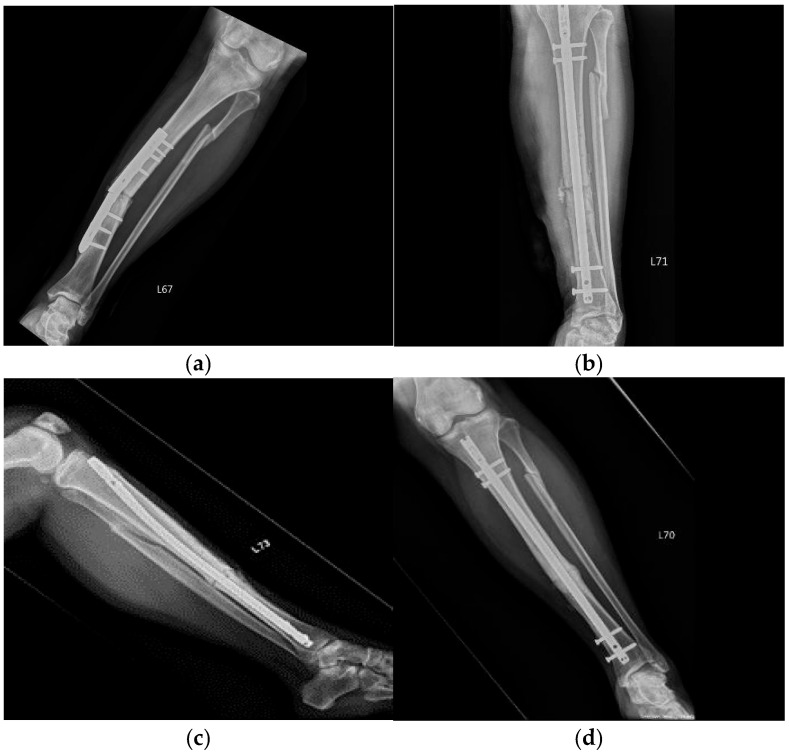
Case 1 is a 50-year-old female who had non-union tibial fracture. (**a**) The failure of plate implant; (**b**) the patient had received revisional intramedullary nail fixation; (**c**) the fracture had not healed 9 months after the revisional intramedullary nailing; (**d**) three months after the 3rd injection of ACS with good union.

**Figure 3 medicina-60-01832-f003:**
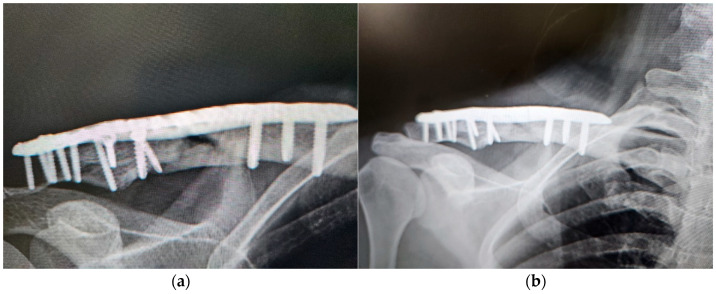
Case 4 is a 39-year-old female treated by ACS due to a non-union clavicle fracture for 9 months after ORIF. (**a**) Before ACS treatment, the defect was visible; (**b**) one month after the 2nd injection. Complete bone healing indicated the excellent efficacy of ACS.

**Figure 4 medicina-60-01832-f004:**
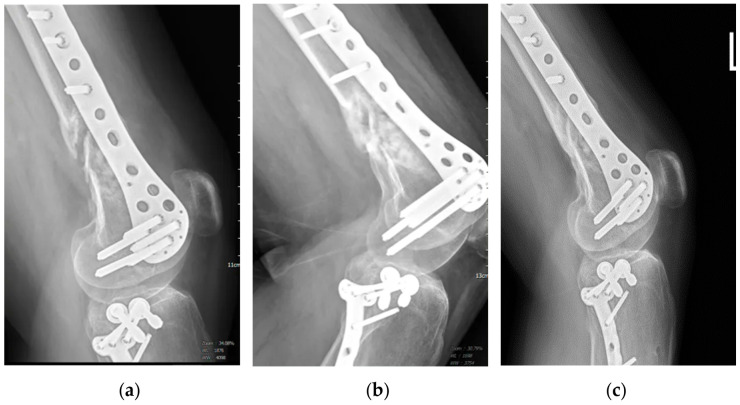
Case 6 is a 34-year-old male treated by ACS due to the non-union femur fracture for 9 months after ORIF with plate fixation. (**a**) Before ACS treatment, the defect was visible; (**b**) one month after the 3rd injection; (**c**) radiographic results one year after the 3rd injection showed complete healing of the bone defect.

**Figure 5 medicina-60-01832-f005:**
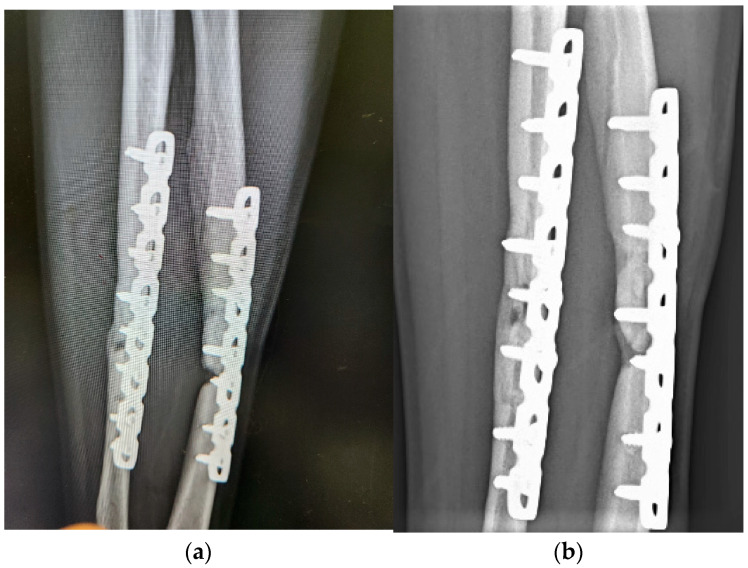
Case 11 is a 27-year-old male who suffered by non-union left radial and ulnar shaft fracture after ORIF. (**a**) Before ACS treatment, obvious defects of both bones were found; (**b**) one month after the 3rd injection by radiographic evaluation.

**Table 1 medicina-60-01832-t001:** Patients’ characteristics.

Case	Age (Years)	Sex	Diagnosis	Previous Treatment Outcomes	Non-Union		ACS Injection				Follow-Up Duration (Months)	Outcomes
					Type	Duration Before ACS (Months)	Times	Guidance	Site	Time to Callus Shown After 1st Injection (Months)		
1	50	Female	Tibial fracture	Plate implant failure, followed by intramedullary nail at 9 months but still non-union	Hypertrophic	9	3	C-arm fluoroscopy	Periosteum	5	12	Union
2	47	Male	Right femoral shaft fracture	Non-union after ORIF	Hypertrophic	9	3	Ultrasound	Intramedullary	4	12	Union
3	45	Male	Right tibia and fibula fracture	Dynamization of distal screws on tibia; ACS injection was given on fibula	Hypertrophic	9	3	C-arm fluoroscopy	Periosteum	4	12	Union
4	39	Female	Right clavicle fracture	Non-union after ORIF	Bone defect	9	2	C-arm fluoroscopy	Intramedullary	3	12	Union
5	66	Male	Right upper femoral fracture	Plate implant failure, followed by gamma nail fixation at 9 months but still non-union	Bone defect	9	3	Ultrasound	Intramedullary	nil	12	Non-union; due to unstable fixation
6	34	Male	Left distal femur fracture	Non-union after ORIF	Bone defect	9	3	C-arm fluoroscopy	Intramedullary	3	12	Union
7	53	Male	Acetabulum bone defect	Non-union after total hip replacement	Bone defect	25	1	C-arm fluoroscopy	Intramedullary	1.5	12	Union
8	50	Male	Left distal femur lateral condyle fracture	Non-union after ORIF	Bone defect	36	2	C-arm fluoroscopy	Intramedullary	3	12	Union
9	39	Male	Right ulnar fracture	Non-union after ORIF	Hypertrophic	14	2	C-arm fluoroscopy	Intramedullary	1.5	12	Union
10	21	Female	Right upper tibia fracture	Non-union after ORIF	Bone defect	12	3	Ultrasound	Intramedullary	4	6	Union
11	27	Male	Left radial and ulnar fracture	Non-union after ORIF	Bone defect	9	3	Ultrasound	Intramedullary	2	6	Union

ACS: autologous conditioned serum; ORIF: open reduction internal fixation.

## Data Availability

Data will be made available on request.

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
