# Peer review of "Effects of Autologous Conditioned Serum on Non-Union After Open Reduction Internal Fixation Failure: A Case Series and Literature Review"

_medicina, 2024, doi:10.3390/medicina60111832_

Round 1
Reviewer 1 Report
Comments and Suggestions for Authors
This study sought to assess the clinical efficacy of ACS injections in patients with non-union fractures following ORIF. This topic is particularly intriguing, given that non-union remains a significant complication even after successful fixation, and reoperation may not always be a feasible option in such cases. The article is well-structured, and despite the limited sample size, the cases included appear to be representative. However, prior to publication, please address the following issues:
Introduction
- Consider expanding the discussion on the current understanding of ACS use in non-union fractures.
- The paragraph on lines 66-70 appears tangential and may not be relevant to the central theme of this article on non-union fractures.
- The objective of the study requires further clarification. The final paragraph is somewhat imprecise and could be better articulated.
Materials and Methods
- Provide additional details about the case involving THR, specifically the location of the bone defect, even though it is mentioned in the results section. Also, include the number of patients involved in the study and the study duration in this section.
- Clarify the criteria used to determine patient eligibility for ACS injections.
- Was the ACS injection administered under ultrasound guidance, or was an alternative method used for administration?
- Please include the ethical committee approval registration number.
Results
- In line 111, you refer to the last follow-up visit. Please specify the exact time frame (e.g., weeks, months). It would be beneficial to know when radiological signs of successful union were first observed following ACS injection. Is there a defined follow-up protocol?
Discussion
- Consider discussing any potential complications associated with the ACS injection, such as the risk of infection.
- Perhaps a short discussion on digital preoperative planning and future perspectives such as robotic surgery or 3D printing would be beneficial to place the paper in a broader context.
- Please consider expanding the discussion on study limitations.
Conclusion
- The current conclusion might seem too optimistic considering the sample size and study design. It would be better if the authors would use a more temperate phrasing and limit the conclusion to the results obtained through the study.
Author Response
Introduction
- Consider expanding the discussion on the current understanding of ACS use in non-union fractures.
Response: Thanks for the comment. To our best knowledge, no study about using ACS injection for patients with non-union fracture after open reduction internal fixation (ORIF). Because PDGF-BB plays an important role in bone formation and regeneration1, and we found that compared to PRP, ACS had a higher level of PDGF-BB to 4.7 times2, the present study was conducted to evaluate the possible utility for non-union fracture. This information is added to the last paragraph of the Introduction, thank you very much. (please see page 2, lines 73-77)
1Caplan, A.I.; Correa, D. PDGF in bone formation and regeneration: new insights into a novel mechanism involving MSCs. J Orthop Res 2011, 29, 1795-1803, doi:10.1002/jor.21462.
2Cheng, P.-G.; Yang, K.D.; Huang, L.-G.; Wang, C.-H.; Ko, W.-S. Comparisons of Cytokines, Growth Factors and Clinical Efficacy between Platelet-Rich Plasma and Autologous Conditioned Serum for Knee Osteoarthritis Management. Biomolecules 2023, 13, 555.
- The paragraph on lines 66-70 appears tangential and may not be relevant to the central theme of this article on non-union fractures.
Response: Thank you for the reviewer's suggestion. The mentioned paragraph has been removed as your suggestion.
- The objective of the study requires further clarification. The final paragraph is somewhat imprecise and could be better articulated.
Response: Thank you for the valuable input. The objective has been rephrased according to your suggestion as follows:’ To our best knowledge, currently no study about using ACS injection for patients with non-union fracture after open reduction internal fixation (ORIF). In this study, we attempted to evaluate the clinical efficacy of ACS injection to explore the possible clinical utility of ACS. The efficacy of ACS on non-union was observed by X-ray evaluation on follow-up visits.’ (please see page 2, lines 73-77)
Materials and Methods
- Provide additional details about the case involving THR, specifically the location of the bone defect, even though it is mentioned in the results section. Also, include the number of patients involved in the study and the study duration in this section.
Response: Thanks for the comment. The patient who underwent THR had an anterosuperior bone defect, which is clarified in the patient section of the Materials and Methods. The X-ray image of this patient is shown below for your review. The study duration and the number of patients included have been added at the end of the patient section in the Materials and Methods. (please see page 2, lines 89-90)

- Clarify the criteria used to determine patient eligibility for ACS injections.
Response: Thank you for the reviewer's suggestion. The inclusion and exclusion criteria are rephrased to clarify in the patient section of the Materials and Methods as follows: ‘Patients aged ≥ 18 years with fracture were eligible to receive the ACS treatment if being radiologically confirmed by certified physicians that still non-union at least 9 months after ORIF. They were asked for consent to participate in this study. In addition, one patient with non-union anterosuperior bone defect after total hip replacement for 25 months was also included. Patients with hypotrophic non-union, fibrosis non-union, unstable fixation, having open wounds, major neurological diseases such as dementia, active thrombovascular diseases, or infections, and who were unco-operative were excluded.’ (please see page 2, lines 81-88)
- Was the ACS injection administered under ultrasound guidance, or was an alternative method used for administration?
Response: Thank you for your advice. Both ultrasound and C-arm fluoroscopy were used for guidance of ACS injection in the study. This information is added to the patient section of the Materials and Methods and the revised Table 1 for each patient.
- Please include the ethical committee approval registration number.
Response: Thank you for the comment. The number is added to the Institutional Review Board Statement. (please see page 11, lines 323-325)
Results
- In line 111, you refer to the last follow-up visit. Please specify the exact time frame (e.g., weeks, months). It would be beneficial to know when radiological signs of successful union were first observed following ACS injection. Is there a defined follow-up protocol?
Response: Thank you for your suggestion. Patients were followed up monthly after the initiation of ACS treatment. Time to the first callus shown on the X-ray images after 1st ACS injection was from 1.5 to 5 months. For the follow-up duration, Case 10 and 11 were followed up for 6 months and the others were followed up for 12 months. This information is added to the Results and the revised Table 1.
Discussion
- Consider discussing any potential complications associated with the ACS injection, such as the risk of infection.
Response: Thank you for the comment. The potential complications with the ACS injection are infection, bleeding, injection pain, nerve injury, or vessel injury. This information is added to the last paragraph of the Discussion. (please see page 10, lines 293-294)
- Perhaps a short discussion on digital preoperative planning and future perspectives such as robotic surgery or 3D printing would be beneficial to place the paper in a broader context.
Response: We deeply thank you for the kind advice, however, since the ACS injection is a relatively simple procedure in clinical practice, robotic surgery or 3D printing might be not necessary. Furthermore, the utility range is speculated to be wider because of the simplicity and ACS injection can be beneficial for more patients who suffer from non-union. This is added to the last paragraph of the Discussion to emphasize the advantage of using ACS injection for non-union.
- Please consider expanding the discussion on study limitations.
Response: Thank you for the advice. The major limitation is the small sample size, further studies with larger populations should be conducted to accumulate clinical experience. In addition, since the blood supply might be limited in the fibrosis tissue, the efficacy in patients with fibrotic non-union is speculated lower, patients may be not suitable for ACS treatment before removing the fibrosis. From the results, the only one who was failed union had unstable fixation, probably ACS treatment is unsuitable for this kind of patients. The limitations are added to the last paragraph of the Discussion. (please see page 10, lines 295-304)
Conclusion
- The current conclusion might seem too optimistic considering the sample size and study design. It would be better if the authors would use a more temperate phrasing and limit the conclusion to the results obtained through the study.
Response: We have rephrased the conclusion according to your kind advice, thank you very much.

Reviewer 2 Report
Comments and Suggestions for Authors
1. It is not classified by type of nonunion. Organize cases according to the classification of hypertrophic nonunion and insert them into the table.
2. Describe the ACS collection in more detail.
3. If you have a fluoroscopy picture of the injection, insert it. Describe in detail how to put it in the center of the nonunion or along the periosteum.
4. Describe setting 3 times to route inspection and setting 2 times to protocol depending on the situation.
1. It is not classified by type of nonunion. Organize cases according to the classification of hypertrophic nonunion and insert them into the table.
2. Describe the ACS collection in more detail.
3. If you have a fluoroscopy picture of the injection, insert it. Describe in detail how to put it in the center of the nonunion or along the periosteum.
4. Describe setting 3 times to route inspection and setting 2 times to protocol depending on the situation.
Author Response
Comments and Suggestions for Authors
1. It is not classified by type of nonunion. Organize cases according to the classification of hypertrophic nonunion and insert them into the table.
Response: Thank you for your suggestion. This information is incorporated into the Results and the revised Table 1.
Describe the ACS collection in more detail.
Response: Thank you for your suggestion. We have added more descriptions of the ACS sterile glass beads including containing tube and the final transfer step in the ACS preparation section.
If you have a fluoroscopy picture of the injection, insert it. Describe in detail how to put it in the center of the nonunion or along the periosteum.
Response: Thank you for the reviewer's suggestion. The pictures of injection guidance of Case 1 are added as new Figure 1 of the revised manuscript, and the injection site is listed in the revised Table 1.
4. Describe setting 3 times to route inspection and setting 2 times to protocol depending on the situation.
Response: We add the description of how to decide the 3rd injection can be canceled at the end of the ACS injection and the follow-up section as follows:’ For example, if the X-ray showed callus one month after the 2nd ACS injection, the scheduled 3rd injection is waived.’ (please see page 3, lines 108-109)

Round 2
Reviewer 1 Report
Comments and Suggestions for Authors
The authors have performed adequate corrections to the issues raised during review. Some large paragraphs, including the last, appear under-referenced and would benefit from some attention regarding proper attribution of intellectual property.
Author Response
Comments and Suggestions for Authors
The authors have performed adequate corrections to the issues raised during the review. Some large paragraphs, including the last, appear under-referenced and would benefit from some attention regarding proper attribution of intellectual property.
Responses to the suggestion
Thank you for your valuable input. We have rewritten the Discussion with attention regarding proper attribution to the citations. For instance, in the last paragraph, we have appropriately discussed and cited the references, describing “The addition of IL-1Ra has been shown to enhance healing of femoral defect in animal experiments [50].”(please see page 5, lines 282-283) , and “Given the fact that the PRP in a gel formulation provided good benefits for patients with delayed union or nonunion [42], further studies to make a gel formulation of ACS for patients with delayed union or nonunion are needed.” (please see page 5, lines 304-306)

Reviewer 2 Report
Comments and Suggestions for Authors
Thank you for correcting the reviewer's requirements well. I found many qualitative improvements.
Author Response
Comments and Suggestions for Authors
Thank you for correcting the reviewer's requirements well. I found many qualitative improvements.
Responses to the suggestion
Thank you for your valuable input.